# A Compositional Heterogeneity Analysis of Mitochondrial Phylogenomics in Chalcidoidea Involving Two Newly Sequenced Mitogenomes of Eupelminae (Hymenoptera: Chalcidoidea)

**DOI:** 10.3390/genes13122340

**Published:** 2022-12-11

**Authors:** Jingtao Jiang, Tong Wu, Jun Deng, Lingfei Peng

**Affiliations:** Biological Control Research Institute, Fujian Agriculture and Forestry University, China Fruit Fly Research and Control Center of FAO/IAEA, Key Laboratory of Biopesticide and Chemical Biology, Ministry of Education, State Key Laboratory of Ecological Pest Control for Fujian and Taiwan Crops, Fuzhou 350002, China

**Keywords:** Eupelmidae, Chalcidoidea, mitochondrial genome, site-heterogeneous mixture model, phylogeny

## Abstract

As next-generation sequencing technology becomes more mature and the cost of sequencing continues to fall, researchers are increasingly using mitochondrial genomes to explore phylogenetic relationships among different groups. In this study, we sequenced and analyzed the complete mitochondrial genomes of *Eupelmus anpingensis* and *Merostenus* sp. We predicted the secondary-structure tRNA genes of these two species and found that 21 of the 22 tRNA genes in *Merostenus* sp. exhibited typical clover-leaf structures, with *trnS1* being the lone exception. In *E. anpingensis*, we found that, in addition to *trnS1*, the secondary structure of *trnE* was also incomplete, with only DHU arms and anticodon loop remaining. In addition, we found that compositional heterogeneity and variable rates of evolution are prevalent in Chalcidoidea. Under the homogeneity model, a Eupelmidae + Encyrtidae sister group relationship was proposed. Different datasets based on the heterogeneity model produced different tree topologies, but all tree topologies contained Chalcididae and Trichogrammatidae in the basal position of the tree. This is the first study to consider the phylogenetic relationships of Chalcidoidea by comparing a heterogeneity model with a homogeneity model.

## 1. Introduction

Eupelmidae (Hymenoptera: Chalcidoidea) includes species that are parasitic and facultatively hyperparasitic on other insects or spiders, with some being natural enemies of many important pests [1]. Worldwide, this family currently includes 49 genera and 1074 species [2], mainly in tropical and subtropical regions [1]. Eupelmidae and its three subfamilies—Calosotinae, Eupelminae, and Neanastatinae—have never been considered as a monophyletic group [3,4]. Eupelmidae, Encyrtidae, and Tanaostigmatidae were once thought to exhibit a relatively close relationship because they all share an expanded acropleuron and jumping ability, among other common characteristics. However, transcriptome-sequence-based phylogenetic analysis has shown that their jumping ability evolved independently on at least three occasions [5,6]. Eupelmidae is more closely related to Pteromalidae than to Encyrtidae and Tanaostigmatidae [5]. A recent study [7] based on 13 protein-coding genes constructed a phylogenetic tree for six families of Chalcidoidea, showing that the pattern of their relationship could be presented as Mymaridae + (Eupelmidae + (Encyrtidae + (Trichogrammatidae + (Pteromalidae + Eulophidae)))).

An insect’s mitochondrial genome is a closed, circular, double-stranded DNA molecule, 14–18 kb in length, which plays an important role in cell metabolism, apoptosis, disease, and aging [8,9,10]. Because of its simple structure and stable genetic material, and in line with the characteristics of maternal inheritance, mitochondrial DNA is widely used for phylogenetic analysis and for genetic differentiation of populations, amongst other study purposes [8,11,12,13]. As of now (13 June 2022), 1,535 complete or nearly complete mitochondrial genome sequences of Hymenoptera, and 23 complete or nearly complete sequences of Chalcidoidea have been published, but only two mitochondrial genome sequences of Eupelmidae have been published thus far (*Anastatus fulloi* was accessed on 5 February 2022; *Eupelmus* sp. was accessed on 18 December 2018), and both of them are incomplete (https://www.ncbi.nlm.nih.gov/).

The mitochondrial genome of insects usually contains 37 genes: 13 protein-coding genes, 22 tRNA genes, 2 rRNA genes, and a non-coding region (CR) [8,14,15]. Wei [16] found that the rate of mitochondrial genome evolution in Hymenoptera was significantly higher than that of other holometabolous orders by calculating the ratio of non-synonymous replacement rate to synonymous replacement rate (Ka/Ks). Insect mitochondrial genomes have high AT content, with varying base compositions and rates of evolution between lineages [17,18,19]. In phylogenetic analysis, the phenomena of base-composition heterogeneity and rapid evolution rates can lead to systematic errors in phylogenetic tree construction [20]. Such phylogenetic trees may be inaccurate if based on the homogeneity model of the mitochondrial genome. To counter such systematic errors, the most common method is to remove the third codon of the protein-coding gene, to reduce the heterogeneity of the sequence. Data partitioning can then be performed, and phylogenetic trees can be constructed using the site-heterogeneity model (CAT-GTR) [18,21,22]. This technique has its drawbacks. Removing the third codon of protein-coding genes may delete important signal and affect-node support [10,23]. However, the CAT-GTR model accommodates data complexity and estimates substitution heterogeneity by calculating a posterior average number of classes [24]. In this study, we analyzed the composition heterogeneity, evolutionary rate, and AT content of the mitochondrial genome of two eupelmid wasps (Hymenoptera, Eupelmidae, Eupelminae) *Eupelmus anpingensis* Masi, and *Merostenus* sp., and reconstructed the phylogeny of Chalcidoidea based on available mitochondrial genome data (including the outgroup, 28 mitochondrial genomes in total). The aim of this study was to analyze the effect of composition heterogeneity on phylogenetic reconstruction based on the mitochondrial genome data of Eupelmidae and to provide new reference material for future phylogenetic studies of Chalcidoidea.

## 2. Materials and Methods

### 2.1. Sample Collection, DNA Extraction and Identification

Two samples were collected from the Tianbaoyan Nature Reserve, Yong’an city, Fujian Province, China, and then stored in pure ethanol at −20 °C until DNA extraction. *Eupelmus anpingensis* was numbered as DNA 817, and *Merostenus* sp. was numbered as DNA 849. The DNA extraction was carried out at the Biological Control Research Institute, Fujian Agriculture and the Forestry University (FAFU).

We extracted genomic DNA from the entire specimen using a DNeasy Blood & Tissue Kit (Qiagen). We followed the manufacturer’s protocol, with some modifications: the specimens were pricked with an insect pin in the abdomen to make a hole and incubated at 56 °C overnight. After DNA extraction, both samples were air dried and mounted on two paper points. The specimens were stored in the FAFU and identified by the corresponding author.

### 2.2. Next-Generation Sequencing, Assembly, and Annotation

We used a total amount of 1.5 µg DNA as input material for DNA sample preparation. We generated sequencing libraries using a Truseq Nano DNA HT Sample Preparation Kit (Illumina, CA, USA) (2 × 150 bp paired-end reads) following the manufacturer’s recommendations. We fragmented the DNA sample using sonication to a size of 350 bp. To ensure the reliability of reads, and to eliminate artificial bias in our subsequent analysis, we first subjected our fast-format raw data to a series of quality control (QC) procedures using in-house C scripts. We used Megahit v1.2.9 [25] and Spades v3.10.1 [26] to assemble the processed data from scratch and eventually obtain a sequence with relatively high coverage. We annotated these sequences by means of the mitos2 online website (http://mitos2.bioinf.uni-leipzig.de/index.py) [27] using the invertebrate mitochondrial genetic code (accessed on 8 May 2022). We verified all tRNA genes again using tRNA-scan [28] and ARWEN [29]. For all protein-coding genes, we searched and verified again using the open reading frame (based on the invertebrate mitochondrial genetic code) in NCBI (https://www.ncbi.nlm.nih.gov/). By such means, we obtained information for a complete mitochondrial genome.

### 2.3. Sequence Analysis

We determined the base composition of these two mitochondrial genomes and the codon usage of protein-coding genes using MEGA7 [30]. We calculated the AT-skew and GC-skew of protein-coding genes and rRNA genes using the following formulas: AT-skew = (A% − T%)/(A% + T%); GC-skew = (G% − C%)/(G% + C%) [31]. We used DnaSP v5 [32] to calculate the ratio of non-synonymous replacement rate (Ka) to synonymous replacement rate (Ks) of protein-coding genes and thus analyze the evolution rate of Chalcidoidea with the mitochondrial genome of *Trichagalma acutissimae* (Cynipoidea) as the outgroup. We also used AliGROOVE v1.08 software [33] to analyze the compositional heterogeneity of both ingroup and outgroup mitochondrial genome sequences.

### 2.4. Phylogenetic Analysis

The two newly sequenced mitochondrial genomes, together with 20 complete mitochondrial sequences of Chalcidoidea and two incomplete mitochondrial sequences of Eupelmidae recorded in GenBank, were taken as the ingroups. The outgroups comprised two species of Proctotrupoidea and two species of Cynipoidea (Table 1). We extracted protein-coding genes and rRNA genes using PhyloSuite v1.2.2 [34]. We compared the protein-coding genes of these 28 species using the G-INS-I algorithm in MAFFT v7 [35] and compared rRNA genes using the Q-INS-I algorithm in MAFFT. We then used Gblock software [36] to select conserved sites. To ensure the quality of the sequences, we manually checked all paired sequences in MEGA 7.

For this study, we set up five datasets, as follows: (1) PCG12 matrix (protein-coding genes with the first and second codon positions of PCGs); (2) PCGs matrix (protein-coding genes with all three codon positions of PCGs); (3) PCG12RNA matrix (protein-coding genes with the first and second codon positions of PCGs and two rRNA genes); (4) PCG123RNA matrix (protein-coding genes with all three codon positions of PCGs and two rRNA genes); and (5) AA matrix (amino acid sequences of PCGs). We concatenated the PCG12RNA matrix and PCG123RNA matrix using PhyloSuite v1.2.2, and extracted the first and second positions of protein-coding genes using DAMBE software [48]. We analyzed the sequence heterogeneity of these five datasets using AliGROOVE v 1.08 with a default sliding window size and aligned the AA matrix using the BLOSUM62 substitution matrix. This metric expressed the pair-wise sequence distance between individual terminals or subclades with terminals outside of the focal group. We recorded and assessed these scoring distances between sequences over the entire data matrix. Metric values ranged from −1 to +1, where −1 indicated distances very different from the average for the entire data matrix, while +1 indicated distances that matched the matrix average.

### 2.5. Construction of Phylogenetic Trees Based on the Site-Homogeneity Model and Site-Heterogeneity Model

For the homogeneity model, we established optimal models of the five datasets by means of PartitionFinder [49], using the greedy search algorithm and Bayesian information criterion (BIC). We constructed the BI tree using MrBayes v3.2.6 [50]. Four simultaneous Markov chains were run for two million generations, with sampling every 10,000 generations, and the burn-in parameter set to 0.25. We constructed the ML tree using IQ-tree [51,52,53] with 1000 ultrafast bootstrapping replicates. The optimal partition schemes and substitution models of the matrix are shown in Appendix A.

For the heterogeneity model, we constructed the phylogenetic tree from the five datasets by means of the CAT-GTR model using PhyloBayes MPI [54]. Two independent trees were searched for, and the process was terminated when the likelihood of the sampled trees had stabilized, and the two runs reached convergence (maxdiff <0.3 and minimum effective size >50). The initial 25% of each run was discarded as burn-in, and a consensus tree was then generated from the remaining trees combined from two runs.

## 3. Results

### 3.1. General Features of the Two Mitochondrial Genomes

The total lengths of the mitochondrial genomes of *E. anpingensis* (GenBank accession number: OP374147) and *Merostenus* sp. (GenBank accession number: OP374146) were 15,479 bp and 16,370 bp, respectively. Both included 37 genes (13 protein-coding genes, 22 tRNA genes, and two rRNAs genes) and a control region (CR), in common with other reported mitochondrial genomes of insects [8,14] as shown in Figure 1 (dorsal and lateral views of these two species are shown in Appendix A). The control region of *E. anpingensis* might not have been completely sequenced, because of the high AT content of the species itself. Comparing these two newly sequenced mitochondrial genomes of Eupelmidae with previously reported Chalcidoidea mitochondrial genomes, *Pteromalus puparum* (Pteromalidae) contained the largest mitochondrial genome (18,217 bp), while *Tetrastichus howardi* (Eulophidae) contained the smallest (14,791 bp). The variation in size of these mitochondrial genomes roughly corresponded to the variable size of the control region. For *E. anpingensis*, we found that 10 protein-coding genes, 16 tRNA genes, and two rRNA genes were located on the majority strand (J-strand), while the remaining genes (three protein-coding genes and six tRNAs) were on the minority strand (N-strand). We also found that the *trnQ* gene of *Merostenus* sp. was located at the N-strand, whereas the *trnQ* gene of *E. anpingensis* was located on the J-strand (Appendix A). The base composition of *E. anpingensis* was A (38.5%), G (10.1%), C (6.7%), and T (44.7%). The base composition of *Merostenus* sp. was A (39.7%), G (11%), C (6.2%), and T (43.1%). The AT content of *E. anpingensis* was 83.3%. For *Merostenus* sp., the AT content was 82.7%. Both mitochondrial genomes thus exhibited a positive GC-skew and a negative AT-skew. Similar findings have been reported for other hymenopteran mitochondrial genomes [7,55].

### 3.2. Protein-Coding Genes and Codon Usage

In *E. anpingensis*, the total length of the 13 protein-coding genes was 11,058 bp, whereas the total length of the protein-coding genes in *Merostenus* sp. was 11,022 bp. Both species exhibited a negative AT-skew in PCGs. The third codon position had the highest AT content, and the second codon position had the lowest AT content (Table 2). All protein-coding genes in these two newly sequenced genomes started with the codon ATN (ATT/ATG/ATA). In *E. anpingensis*, the terminal codons were TAA or TAG, whereas in *Merostenus* sp., the terminal codon was always TAA. Figure 2 shows the relative synonymous codon usage (RSCU) in the genomes of the two newly sequenced species and those of two other Eupelmidae species (downloaded from GenBank). For all four species, codon usage of protein-coding genes is basically the same, with third codon positions more likely to be A or T than G or C. The most frequently used codons are UUA (*Leu2*), AUU (*Ile*) and UUU (*Phe*), which are all composed of just A or U. This shows that the codons of protein-coding genes of these four species prefer to use A and U in the third position, which explains the high AT content in the sequences overall. In addition, the Ka/Ks ratio revealed, using DnaSP v5, that the entire Eupelmidae family exhibits a high rate of evolution.

### 3.3. tRNA Genes and rRNA Genes

All the tRNA genes of *E. anpingensis* and *Merostenus* sp. can be identified through the mitos2 website, and these can be verified again through ARWEN and tRNA-scan, so that the secondary structures of all tRNA genes can be obtained (Figure 3 and Figure 4). In *Merostenus* sp., the secondary structures in 21 of the 22 tRNA genes are typical clover-leaf structures. The exception is *trnS1*, which has lost the DHU arm. In *E. anpingensis*, *trnS1* and *trnC* have both lost the DHU arm and hence do not exhibit the clover-leaf structure. We also found serious structural loss in the *trnE* secondary structure with only an anticodon loop and DHU arm remaining. Previous research papers have reported similar losses of tRNA structures [10,56]. In addition to the changes in the secondary structure, we also found mismatches in the tRNA bases. In both *Merostenus* sp. and *E. anpingensis*, we identified five base mismatches, all of which were G–U mismatches, which are common in Hymenoptera [55,57,58].

rRNA genes (*rrnL* and *rrnS*) are commonly located at *trnL1-trnV* and *trnV*-control regions [9]. However, in *E. anpingensis*, we found that *rrnS* and *rrnL* genes were located at *trnV-trnA* and *trnQ-trnL1*. In *Merostenus* sp., *rrnS* and *rrnL* genes were located at *trnV-trnQ* and *trnA-trnL1*. The total lengths of rRNA genes in *E. anpingensis* and *Merostenus* sp. were 1936 bp and 2064 bp, respectively. The *rrnS* and *rrnL* genes in the newly sequenced genomes of these two species both exhibited a negative AT-skew and a positive GC-skew (Table 2).

### 3.4. Phylogenetic Tree Based on Homogeneity Model

We established optimal models based on the ML tree and the BI tree for all five datasets using PartitionFinder. For the AA dataset, the optimal model was MTREV + I + G; for each of the other four datasets, the optimal model was GTR + I + G. The result of the BI-tree results for the three datasets (PCGs, PCG12RNA, and PCG123RNA) had the same topological structure. We found different tree topologies for the AA dataset based on the BI analysis and ML analysis (Appendix A). In this case, Trichogrammatidae was the sister group of Eulophidae + Pteromalidae, and Encyrtidae was closer to Eupelmidae, Trichogrammatidae, Eulophidae and Pteromalidae, as Chalcididae + (Encyrtidae + (Eupelmidae + (Trichogrammatidae + (Eulophidae + Pteromalidae)))), indicating that Encyrtidae has an earlier origin than Trichogrammatidae. This finding is inconsistent with the results of Heraty et al. [4] and Peters [5]. However, the phylogenetic tree based on our BI analysis and ML analysis of the PCG12 dataset exhibited a topological structure almost identical to the three datasets previously described, namely, PCGs, PCG12RNA, and PCG123RNA, as shown in Appendix A. For this reason, we suggest that the reliability of phylogenetic trees constructed using AA datasets should be carefully considered in future studies. For the present, we state that our BI-tree analysis recovered Chalcididae + (Trichogrammatidae + (Pteromalidae + Eulophidae) + (Eupelmidae + Encyrtidae)), and the ML tree constructed using IQ-tree methods produced an almost identical topology (Figure 5).

### 3.5. Phylogenetic Tree Based on Heterogeneity Model

We analyzed the heterogeneity of our five datasets using AliGROOVE software and found that all datasets exhibited various degrees of heterogeneity; in particular, those datasets containing the third codon position of protein-coding genes (Figure 6 and Appendix A). The PCGs datasets exhibited higher degrees of heterogeneity than the AA datasets, and the third codon positions of PCGs exhibited a distinctly higher heterogeneity than the first and second positions. We calculated Ka/Ks values for each taxon (Figure 7); for all families, these values were less than 1, indicating a negative selection in the evolution of the genes used (PCGs). We found that AT content exhibited a similar tendency to Ka/Ks values. The sequences of Eulophidae showed a comparatively higher AT content than other families, whereas among Chalcidoidea, Encyrtidae showed a lower AT content.

For all five datasets, we constructed a BI tree using the CAT-GTR model. Our results showed that different datasets indicated various topologies (Figure 8, Figure 9, and Appendix A). Nearly identical topological trees were generated in the datasets based on PCG12 and PCGs, as Chalcididae + (Trichogrammatidae + (Pteromalidae + (Eulophidae + (Eupelmidae + Encyrtidae)))) (Figure 9). However, in the PCG12 datasets, one species of Trichogrammatidae, *Megaphragma amalphitanum*, did not group with other Trichogrammatidae, possibly due to a lack of informative loci (Appendix A). The datasets based on PCG12RNA and PCG123RNA formed a consistent topological tree that was identical to the trees part constructed by the first two datasets. Both placed Chalcididae and Trichogrammatidae near the base of the entire tree (Figure 8), as Chalcididae + (Trichogrammatidae + (Pteromalidae + (Eupelmidae + (Eulophidae + Encyrtidae)))). The topological tree obtained from the AA dataset based on the CAT-GTR model was Chalcididae + Trichogrammatidae + (Pteromalidae + Eulophidae) + (Eupelmidae + Encyrtidae), which was consistent with the homogeneity-model tree topology of the other three datasets except for the AA and PCG12 matrices.

### 3.6. Comparative Analysis Based on Homogeneity Model and Heterogeneity Model

In summary, with the exception of the BI trees and ML trees constructed from the AA datasets under the homogenous model that did not place Trichogrammatidae closer to the base, the topological trees constructed from our datasets suggested that Chalcididae and Trichogrammatidae may have an earlier origin, a finding consistent with previous studies [3,43]. Comparing the results obtained from our homogeneity and heterogeneity models, we found some inconsistency concerning the classification status of Eupelmidae. Based on the homogeneity model, all datasets were consistent, indicating that Eupelmidae was closer to the end of the tree. However, based on the heterogeneity model, the PCG123RNA, PCG12RNA, PCGs, and PCG12 matrices did produce a definitive classification status for Pteromalidae, Eupelmidae, and Eulophidae. When the third codon position was removed, the influence of heterogeneity on tree construction was reduced, but different datasets did not always produce the same topological structure. These findings highlight the need for better means of constructing more reliable phylogenetic trees.

## 4. Conclusions

In this study, we sequenced the mitochondrial genomes of two different genera of Eupelmidae, *E. anpingensis* and *Merostenus* sp. Through the calculation of AT content, Ka/Ks analysis of the two species, and the heterogeneity testing of different datasets using AliGROOVE software, we found high heterogeneity in both the ingroup and outgroup. The constructed ML tree and BI trees based on the homogeneity model and the constructed BI trees based on the heterogeneity model did not produce consistent results concerning the topological structures of the six selected families, and this may have been the result of high AT content, rapid evolutionary rate, or high sequence heterogeneity. Increasing the sample size may reduce systematic error; however, increasing the number of ingroup taxa in the phylogenetic analysis will make it harder to evaluate the correct phylogenetic tree [20]. Overall, this is the first attempt to study the phylogeny of Chalcidoidea by comparing a heterogeneity model with a homogeneity model. Our findings provide reference material for further research on Chalcidoidea.

## Figures and Tables

**Figure 1 genes-13-02340-f001:**
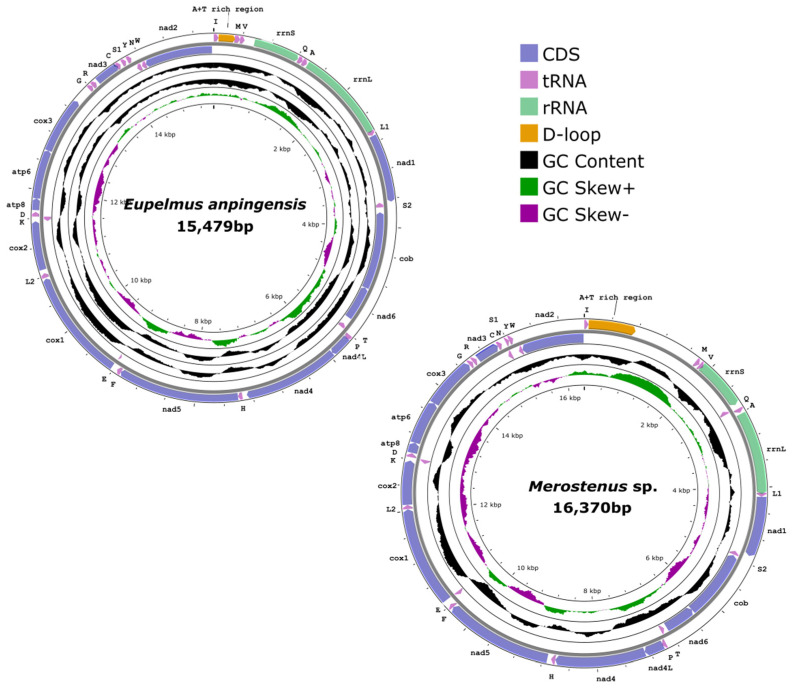
Mitochondrial genomes of *Eupelmus anpingensis* and *Merostenus* sp.

**Figure 2 genes-13-02340-f002:**
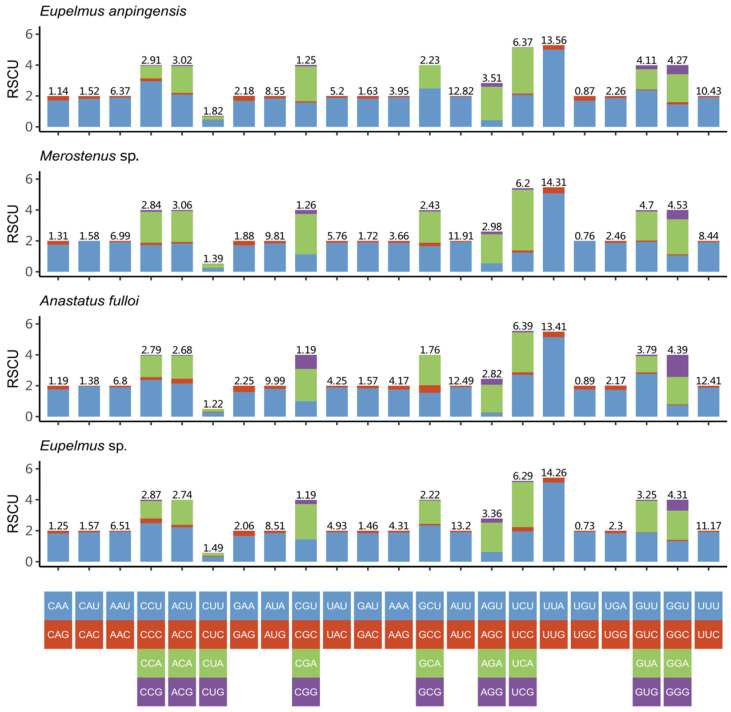
Relative synonymous codon usage (RSCU) of four species of eupelmid wasps.

**Figure 3 genes-13-02340-f003:**
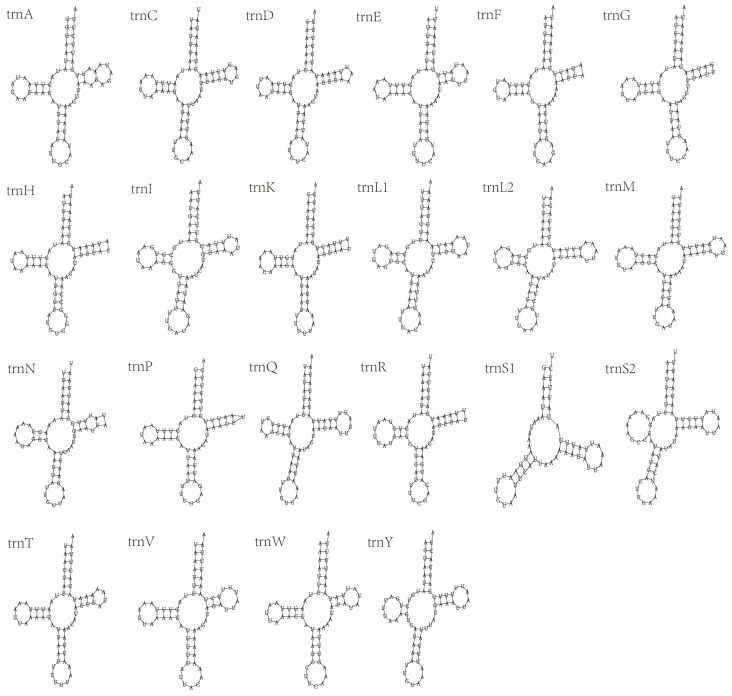
Secondary structures of tRNA genes of *Merostenus* sp.

**Figure 4 genes-13-02340-f004:**
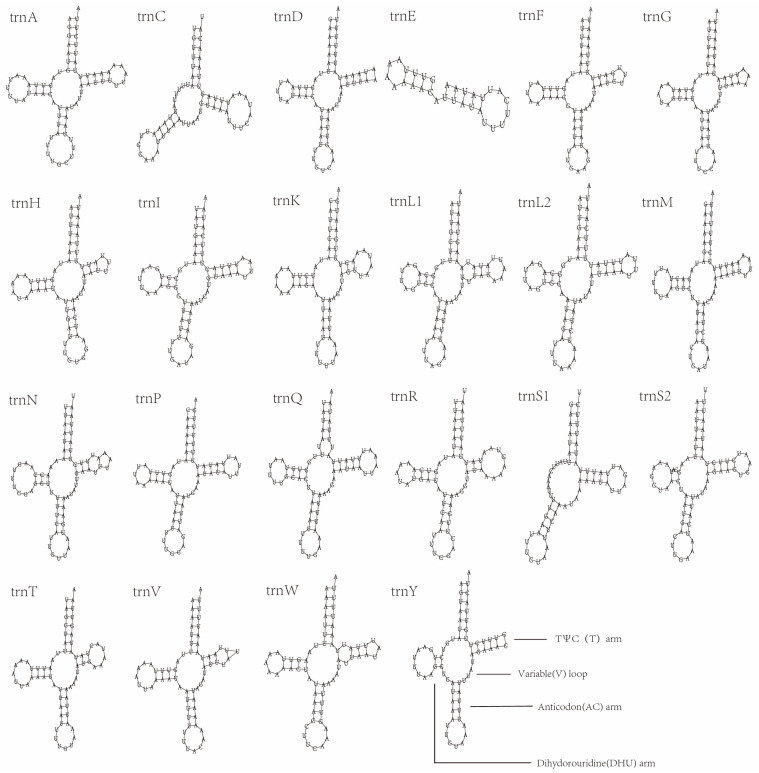
Secondary structures of tRNA genes of *Eupelmus anpingensis*.

**Figure 5 genes-13-02340-f005:**
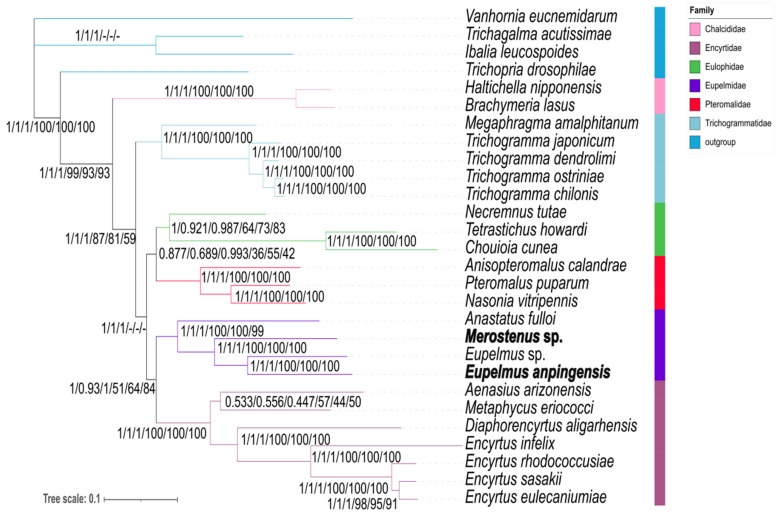
Phylogenetic tree inferred from MrBayes and IQ trees based on the datasets of PCGs, PCG12RNA, and PCG123RNA. Supports at nodes (from left to right) are Bayesian posterior probabilities (PPs) for PCGs, PCG12RNA, and PCG123RNA, and ML bootstrap support values (BSs) for PCG12RNA, PCG123RNA, and PCGs.

**Figure 6 genes-13-02340-f006:**
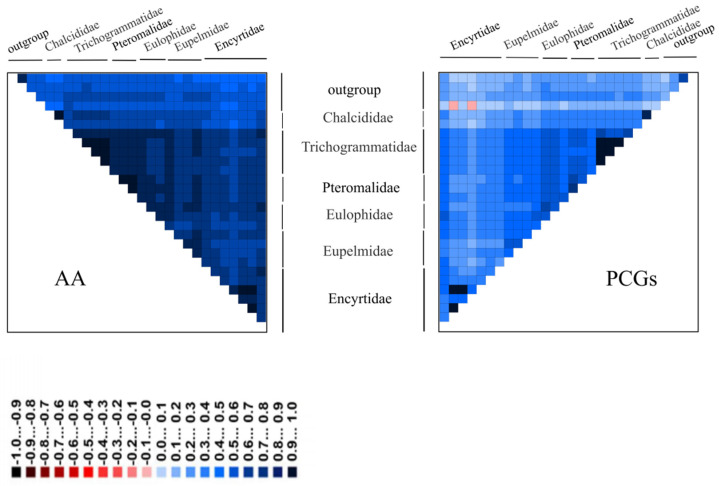
Heterogeneous analysis of 28 species (including four outgroups) based on AA and PCG datasets. The mean similarity score between sequences is represented by a colored square. AliGROOVE scores range from −1, which indicates distances very different from the average for the entire data matrix (red color), to +1, which indicates distances matching the matrix average (blue color).

**Figure 7 genes-13-02340-f007:**
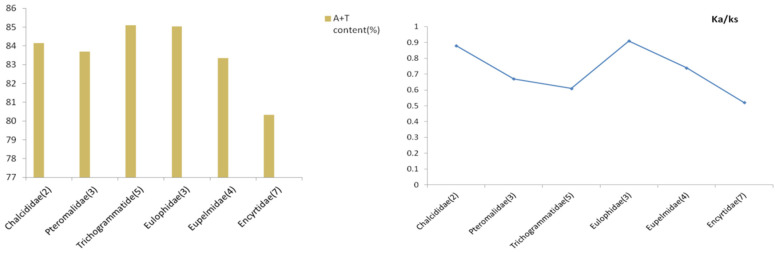
AT content (left) and Ka/Ks values (right) based on PCGs for different families.

**Figure 8 genes-13-02340-f008:**
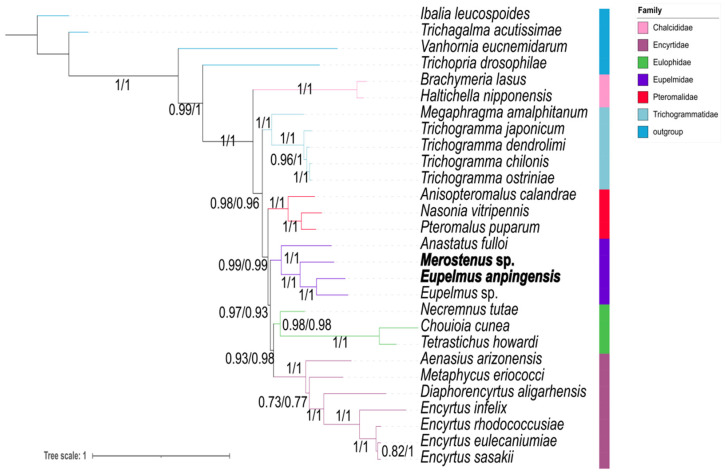
Phylogenetic tree inferred from PhyloBayes based on PCG12RNA (**left**) and PCG123RNA (**right**) datasets. Supports at nodes are Bayesian posterior probabilities (PPs).

**Figure 9 genes-13-02340-f009:**
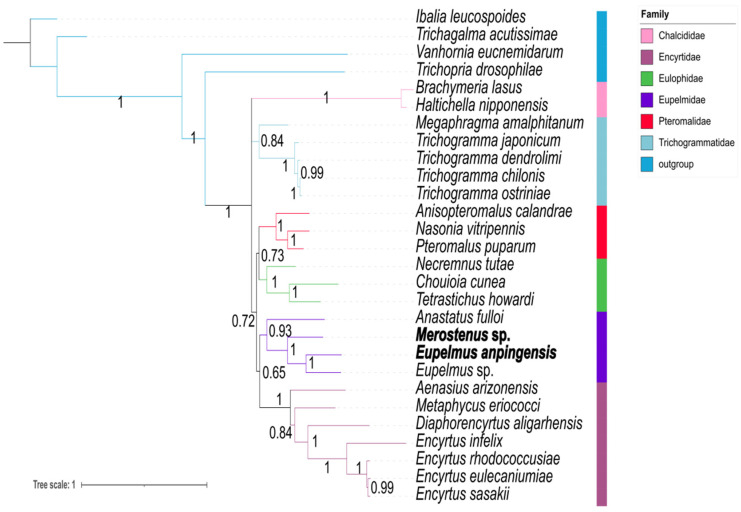
Phylogenetic tree inferred from PhyloBayes based on AA dataset. Supports at nodes are Bayesian posterior probabilities (PPs).

**Table 1 genes-13-02340-t001:** List of mitochondrial genomes used for phylogenetic analysis. (★ indicates the two species sequenced in this study).

Subfamily	Species Name	Accession Number	Length (bp)	A + T (%)	Reference
Pteromalidae	*Anisopteromalus calandrae*	MW817149	15,954	82.9	[37]
	*Nasonia vitripennis*	MT985330	15,291	83.5	Unpublished
	*Pteromalus puparum*	NC_039656	18,217	84.7	Unpublished
Eulophidae	*Tetrastichus howardi*	MZ334468	14,791	85.5	[38]
	*Chouioia cunea*	MW192646	14,930	85.1	[39]
	*Necremnus tutae*	NC_053857	15,252	84.5	[40]
Trichogrammatidae	*Trichogramma dendrolimi*	KU836507	16,878	84.7	Unpublished
	*Trichogramma ostriniae*	NC_039535	16,472	85.4	[41]
	*Megaphragma amalphitanu*	NC_028196	15,041	85.3	[42]
	*Trichogramma japonicum*	NC_039534	15,962	84.9	[41]
	*Trichogramma chilonis*	MT712144	16,176	85.2	Unpublished
Chalcididae	*Brachymeria lasus*	MZ615567	15,147	84.5	[43]
	*Haltichella nipponensis*	MZ615568	15,334	83.8	[43]
Encyrtidae	*Encyrtus rhodococcusiae*	NC_051460	15,694	79.1	Unpublished
	*Encyrtus eulecaniumiae*	NC_051459	15,692	80	Unpublished
	*Encyrtus sasakii*	NC_051458	15,708	79.2	Unpublished
	*Aenasius arizonensis*	NC_045852	15,373	79.6	[44]
	*Encyrtus infelix*	NC_041176	15,698	78.4	[45]
	*Diaphorencyrtus aligarhensis*	NC_046058	16,264	81.8	[46]
	*Metaphycus eriococci*	NC_056349	15,749	84.2	Unpublished
Eupelmidae	*Anastatus fulloi*	OK545741	15,692	83.9	[7]
	*Eupelmus* sp.	MG923493	17,037	83.5	[47]
	*Merostenus* sp. ★	OP374146	16,370	82.7	This study
	*Eupelmus anpingensis* ★	OP374147	15,479	83.3	This study

**Table 2 genes-13-02340-t002:** Base composition of each position of protein-coding genes and rRNA genes.

Merostenus sp.									
Regions	Size (bp)	T(U)%	C%	A%	G%	A + T%	G + C%	AT-skew	GC-skew
Full genomes	16,370	43	6.2	39.7	11	82.7	17.3	−0.040	0.277
PCGs	11,022	44.7	8.3	36.5	10.4	81.3	18.7	−0.101	0.112
1st codon position	3674	38.2	8.4	38.3	15.2	76.5	23.5	0.001	0.289
2nd codon position	3674	50.4	14.5	23.2	11.9	73.6	26.4	−0.370	−0.098
3rd codon position	3674	45.7	2.2	48.1	4.0	93.8	6.2	0.026	0.290
rrnS	775	43.4	4.5	42.7	9.4	86.1	13.9	−0.008	0.353
rrnL	1289	42.7	5.0	42.8	9.4	85.5	14.5	0.001	0.303

Eupelmus anpingensis									
Regions	Size (bp)	T(U)%	C%	A%	G%	A + T%	G + C%	AT-skew	GC-skew
Full genomes	15,479	44.8	6.7	38.5	10.1	83.3	16.7	−0.076	0.204
PCGs	11,058	46.9	8.5	34.5	10.1	81.4	18.6	−0.152	0.086
1st codon position	3686	38.9	8.6	38.1	14.4	77.0	23.0	−0.010	0.252
2nd codon position	3686	51.1	14.5	22.3	12.1	73.4	26.6	−0.392	0.090
3rd codon position	3686	50.8	2.4	43.1	3.7	93.9	6.1	−0.082	0.213
rrnS	644	43.3	5.6	41.5	9.6	84.4	15.2	−0.021	0.263
rrnL	1292	45.0	5.1	40.1	9.8	85.1	14.9	−0.058	0.315

## Data Availability

The raw sequencing data of *E. anpingensis* and *Merostenus* sp. were deposited in the NCBI Sequence Read Archive (SRA) database, as follows: *E. anpingensis* (PRJNA891614); *Merostenus* sp. (PRJNA891649).

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
