# Peer review of "A Compositional Heterogeneity Analysis of Mitochondrial Phylogenomics in Chalcidoidea Involving Two Newly Sequenced Mitogenomes of Eupelminae (Hymenoptera: Chalcidoidea)"

_genes, 2022, doi:10.3390/genes13122340_

Round 1

Reviewer 1 Report

The manuscript presents interesting results that contribute to the knowledge of the evolutionary relationship of the Chalcidoidea superfamily. However, I CANNOT CONSIDER THIS MANUSCRIPT SUITABLE FOR PUBLICATION until the authors make the taxonomic identification of the taxa that had the mitochondrial genome described (IDENTIFYING THE SPECIES THEY REPRESENT). Publish the data as Merostenus sp. and Eupelmus sp. does not allow advancing knowledge about the group.

In addition, I ask the authors to improve the resolution of the images, with an emphasis on phylogenetic reconstructions, since it is not possible to read the scientific names.

Author Response

Dear Reviewer

Thank you very much for your suggestions. We identified the specimens very carefully, and we think that one is Eupelmus anpingensis Masi, 1927, and the other one Merostenus sp. maybe a new species, and till now only one specimen in our collection. So, we think we would bettle still use Merostenus sp. for its name.  The images of phylogenetic tree also have improved.

Best Regards

Sincerely Yours,

Peng Lingfei E-mail: [email protected]

2022/10/21

Reviewer 2 Report

The article is interesting to be published because it tests the phylogenetic relationships in the superfamily Chalcidoidea based on the heterogeneity model through mitochondrial genomes. The article does adhere to the journal`s standards. The research question is outlined and all the key elements are present, namely: title, abstract, introduction, methodology, results, and conclusions. The title clearly reflects the article contest as also does the abstract. "Introduction" deeply describes what the authors hoped to achieve; accurately and clearly states the problem being investigated. They accurately explain how the data was gathered; the design is suitable for answering the question posed. Results are clearly laid out and are presented in a logical sequence. When considering the whole article, the balance of the illustrative material to the text is sufficient; the figures are an important part of the story. So, this study would have a good contribution to deepening our understanding of phylogenetic relationships within Chalcidoidea. 

Some critical remarks. In the introduction part (lines 27-28) I find inappropriate self-citation (source N1); there are many other suitable sources to be quoted here. Again, in the same chapter (lines 44-47) the statement about the number of up-to-date mitochondrial sequences in Hymenoptera must be supported by a source where this information comes from.

Author Response

Dear Reviewer

Thank you very much for your suggestions. We edite the self-citation carefully,and also add the link of NCBI to support the number of up-to-date mitochondrial sequences in Hymenoptera.

Best Regards

Sincerely Yours,

Peng Lingfei E-mail: [email protected]

2022/10/21

Round 2

Reviewer 1 Report

Small modifications:

Line 3 and 4

Change “Eupelminae (Hymenoptera: Chalcidoidea: Eupelmidae)” for “Eupelminae (Hymenoptera: Chalcidoidea)”

Line 16

Change “Eupelmus anpingensis” for “E. anpingensis

Line 67

Change “…of two eupelmid wasps” for “…of two eupelmid wasps Eupelmus anpingensis Masi, 1927 (Hymenoptera, Eupelmidae) and Merostenus sp.”

Line 73

Include the number of specimens used.

Lines 169-184

Include the author and year of description in all scientific names cited for the first time in the text.

Lines 262, 307, 313

The phylogeny is extremely low resolution (I can barely read the scientific names)

Author Response

Dear Reviewer

Thank you for your nice suggestions, we have studied comments carefully and have made correction which we hope meet with approval. Revised portion are marked red in paper.

Mainly two points are following: 1. We added the related references of specimens we used in this study, but some of these are not published; 2. The phylogeny map were revised, getting more clear.

We tried our best to improve the manuscript and made some changes in the manuscript. These changes will not influence the content and framework of the paper. And here we did not list the changes but marked in red in revised paper. We appreciate for your warm work earnestly, and hope that correction will meet with approval.

Once again, thank you very much for your comments and suggestions.

Best Regards

Jingtao Jiang